# Application of co-culture technology of epithelial type cells and mesenchymal type cells using nanopatterned structures

**Taek-Hee Jung**[1☯], **Eun-Bin Chung**[2☯], **Hyung Woo Kim**[3☯], **Seong Woo Choi**[4☯], **Soon-Jung Park**[1], **Anthony Safaa Mukhtar**[1], **Hyung-Min Chung**[2], **Eunmi Kim**[5], **Kang Moo Huh**[6], **Dong Sung Kim**[3‡]*, **Sun-Woong Kang**[7,8‡]*, **Sung-Hwan Moon**[1,9‡]*

**1** Research Institute, T&R Biofab Co. Ltd, Siheung, Republic of Korea, **2** Department of Stem Cell Biology, School of Medicine, Konkuk University, Seoul, Republic of Korea, **3** Department of Mechanical Engineering, Pohang University of Science and Technology, (POSTECH) Pohang, Pohang, Republic of Korea, **4** Ischemic/Hypoxic Disease Institute, Seoul National University College of Medicine, Seoul, Republic of Korea, **5** R&D Unit, Amorepacific Corporation, Yongin-si, Gyeonggi-do, Republic of Korea, **6** Department of Polymer Science and Engineering Chungnam National University, Daejeon, Republic of Korea, **7** Research Group for Biomimetic Advanced Technology, Korea Institute of Toxicology, Daejeon, Republic of Korea, **8** Department of Human and Environmental Toxicology, University of Science and Technology, Daejeon, Republic of Korea, **9** Department of Medical Science, School of Medicine, Konkuk University, Seoul, Republic of Korea

☯ These authors contributed equally to this work.
‡ These authors also contributed equally to this work.
* smkds@postech.ac.kr (DSK); swkang@kitox.re.kr (SWK); sunghwanmoon@kku.ac.kr (SHM)

**Data Availability Statement:** All relevant data are within the paper and its Supporting Information files.

## Abstract

Various nanopatterning techniques have been developed to improve cell proliferation and differentiation efficiency. As we previously reported, nanopillars and pores are able to sustain human pluripotent stem cells and differentiate pancreatic cells. From this, the nanoscale patterns would be effective environment for the co-culturing of epithelial and mesenchymal cell types. Interestingly, the nanopatterning selectively reduced the proliferative rate of mesenchymal cells while increasing the expression of adhesion protein in epithelial type cells. Additionally, co-cultured cells on the nanopatterning were not negatively affected in terms of cell function metabolic ability or cell survival. This is in contrast to conventional co-culturing methods such as ultraviolet or chemical treatments. The nanopatterning appears to be an effective environment for mesenchymal co-cultures with typically low proliferative rates cells such as astrocytes, neurons, melanocytes, and fibroblasts without using potentially damaging treatments.

## Introduction

It is important to provide conditions similar to the complex and intricate environment of the body for the proliferation and differentiation of cells [1–3]. Basically, in order to regulate cell behavior *in vitro*, soluble factors (growth factors, cytokines, small molecules, etc.) that mimic the microenvironment of native tissues are used. However, evidence has been reported that in addition to the biochemical environment, the physical environment in contact with the cell plays an important role in the activity of the cells [4–8].

**Funding:** This research was supported by the Bio & Medical Technology Development Program of the National Research Foundation funded by the MSIP (NRF-2016M3A9B4919616, NRF-2019M3A9H1103331) and by a grant (20000325) from the Technology Innovation Program funded from the Ministry of Trade, Industry and Energy (MOTIE), and National Research Foundation grant (MSIT) (No. 2017R1A2A1A05001090) funded by the Republic of Korea government. T&R Biofab Co. Ltd and Amorepacific Corporation did not provide financial support, and did not have any roles in this study. The authors belonging to T&R Biofab and Amorepacific companies recently moved to the companies, and no funds have been contributed by the companies.

**Competing interests:** The authors have declared that no competing interests exist. T&R Biofab Co. Ltd and Amorepacific Corporation do not alter our adherence to PLOS ONE policies on sharing data and materials.

The basement membrane, which is a physical environment, is an extracellular matrix (ECM) that provides a substrate for survival, proliferation, and differentiation of cells. Recently, various pattern structures, graphene, peptides, and the like have been developed to mimic the ECM conditions [7, 9–11]. In particular, in the case of the pattern structure, there is an advantage that can be produced in an artificially uniform form [12, 13]. In addition, topographic studies of the basement membrane reveal the presence of pores of various sizes and the presence of nano-sized elevations, which are produced by intertwining fibrous proteins such as laminin, fibronectin, and collagen [14, 15]. For these reasons, it has been suggested that cell behavior can be closely influenced by various nanoscale features. Nanotechnology, such as self-assembly and lithography, has been developed that can implement various types of nanosurfaces to optimize cell-nanographic interactions for enhancing cell behavior and activity. For example, polydimethylsiloxane nanopatterned with line gratings promote differentiation of human mesenchymal stem cells (MSCs) into neuronal lineage cells [16, 17], and osteogenic differentiation of human MSC cells were induced in disordered arrays of nanopites [18]. In addition, human MSCs cultured on nanoporous polystyrene surfaces have improved adipogenesis compared to the control flat surface, whereas osteogenesis of human MSCs was induced on nanopillar-shaped polystyrene surfaces [19].

Previously, we fabricated various sized nanopattern pillars and optimized for effective pattern size for maintain of undifferentiated human pluripotent stem cells (hPSCs) [12]. Also recently we reported that nanopatterned pore type dishes are effective for the differentiation of pancreatic cells [13]. Interestingly, the results of previous studies show that MSC cells, which are mesenchymal type cells in nanopatterned structures, are more effective for differentiation than proliferation, whereas undifferentiated hPSC cells, which are epithelial type cells, promote proliferation and differentiation into pancreatic cells of epithelial type. Therefore, we thought that mesenchymal type cells and epithelial type cells, which are classified according to cell types, may have different growth rates in nanopatterns. In this study, we compared the proliferative capacity between fibroblasts of mesenchymal type and keratinocytes of epithelial type in a nanopore structure, and we found that the nanopattern can be used as a new technique to co-culture mesenchymal type cells and epithelial type cells.

## Materials and methods

### Cell culture

In this study, we used the previously reported nanopatterned dishes [13, 20]. Nanopattern dishes and flat cell culture dishes were coated with 0.1% gelatin for 20 min at room temperature before cell seeding. Keratinocytes, fibroblasts were cultured in DMEM with 10% FBS, 1% NEAA and 1% penicillin-streptomycin. Undifferentiated hPSCs (H9-hESC lines) were grown on proliferative mouse fibroblasts and MMC-treated mouse fibroblasts in DMEM/F12 with 20% knock-out serum replacement, 100 mM β-mercaptoethanol, 1 mM L-glutamine, 1% NEAA, 1% penicillin-streptomycin and 4 ng/ml bFGF. The MMC was treated on fibroblasts at 10 μg/ml concentration. All movies were taken with a Lumasope 720 microscope (Etaluma) [21]. All materials except DMEM (Sigma. USA) and bFGF (R&D systems, USA) were purchased from Gibco. The study was approved by the institutional review board (IRB) of the Konkuk University (KUH1280080).

### Immunocytochemistry

Cells were washed with PBS and fixed with 4% v/v paraformaldehyde for 20 min at 4˚C. Then, they were permeabilized with 0.03% Triton X-100 for 15 min and blocking solution for 30 min. Cells were incubated with primary antibody Oct4 (1:200), E-cadherin (1:200) (Abcam,

UK). The following day the cells were incubated with Alexa Flour 488- or 594- conjugated secondary antibodies (1:1000) (Invitrogen, USA). Finally, the cells were stained with DAPI (1:700) (Invitrogen). All images were acquired using a fluorescence microscope (Nikon TE2000-U, Japan).

## Alkaline-Phosphatase (AP) staining

AP staining was performed using Alkaline Phosphatase Staining Kit II (STEMGENT, USA) following the manufacturer's instructions. Cells were fixed with fixing solution for 2 to 5 minutes at room temperature and then washed with PBS. After fixation, AP substrate solution was added to the culture dish and the cells were incubated in a dark at room temperature for 5 to 15 minutes.

## CCK-8 assay

The analysis of cell proliferation was performed at 24, 48 and 72 hours. Viable cells were identified using a cell counting kit-8 (CCK-8, Japan) assay. The spectrophotometric absorbance was measured with a microplate reader. Experiments were performed in triplicate.

## Quantitative real-time PCR and reverse transcription PCR

Total RNA was isolated using TRIzol reagent (Invitrogen) according to the manufacturer's instructions. RNA concentration was measured using NanoDrop One$^c$ Microvolume UV-Vis Spectrophotometer (Thermo Scientific, USA), and cDNA was synthesized by High Capacity cDNA Reverse Transcription Kit (Applied Biosystems, USA) according to the manufacturer's manual. Reverse transcription PCR was performed that the synthesized cDNA, gene-specific primer with the mixture were size fractionated by 1% agarose gel electrophoresis and visualized by ethidium bromide staining. The final analysis was obtained using an image analyzer (Bio-Rad, Hercules, CA, USA). The primer sequences are listed in S1 and S2 Tables.

## Quantitative PCR and RT2 profiler PCR array

Cultured cells on flat and nanopatterns was analyzed via human epithelial-mesenchymal transition (EMT) and mouse cell cycle RT$_2$ Profiler PCR Array kit (Qiagen, Germany). The PCR arrays were performed using SYBR Green expression assays (Roche, Germany) on LightCycler® 96 Real-Time PCR System machine (Roche, Germany) following the manufacturer's instructions. Total RNA isolation and cDNA synthesis were performed same way on 2.5 material methods. Data were normalized to ACTB, B2M, GAPDH, HPRT1 and RPLP0 of housekeeping genes. Data analysis was shown using Data Analysis Center of Qiagen.

## Epithelial-Mesenchymal Transition (EMT) array

Upon the cultivation of keratinocyte and fibroblast on nanopore pattern, the expression of 84 genes associated with the epithelial-mesenchymal transition (EMT) analyzed via EMT RT2 Profiler PCR Array kit (Qiagen). Target genes were screened using a LightCycler® 96 Real-Time PCR System (F. Hoffmann-La Roche Ltd., Switzerland) following the manufacturer's instructions. cDNA was synthesized using the High-Capacity cDNA Reverse Transcription Kit (Thermo Scientific). Data analysis was performed using SA Biosciences and Cluster software (RT2 profiler 3.5, Qiagen).

### FACS analysis of cell cycle

Collected cells were fixed with 70% ethanol 4–8 hours at 4˚C, then stained with FxCycle PI/RNase staining solution (Molecular Probes, USA). Cellular DNA contents were acquired using SONY SH800S Cell Sorter (Sony Biotechnology, Japan).

### XFe$^{96}$ metabolism analyzer

Oxidative and glycolysis phosphorylation flux analysis was performed using kits from Seahorse Biosciences. Cells were seeded on XFe96 cell culture microplates (Seahorse Biosciences, USA) and grown to 70–90% confluence prior to analysis. Culture media were changed to XF base medium (Seahorse Biosciences), supplemented with 5 g/L glucose and 2 mM pyruvate for the oxidative phosphorylation assay. Before the assay, plates were transferred into a non-$CO_2$ incubator at 37˚C and maintained for 1 hours. To quantify the oxygen consumption rates, we first measured basal oxygen consumption rates, followed by a series of changes in oxygen consumption when cells were sequentially treated with oligomycin, FCCP, and a combination of rotenone and antimycin A.

### Cytokine and chemokine array

Cells were seeded onto ten 100 mm cell-culture dishes and each dish was cultured to 80–90% confluence. Half of the samples were then treated with MMC to represent cytokine release from the MMC-treated experimental group. All cells were washed with PBS and the medium was changed to high glucose DMEM (Lonza, Switzerland). After 5 days of culture, the supernatants were collected and analyzed by using a biotin label-based mouse cytokine antibody array for 144 mouse cytokines (RayBiotech, USA) in accordance with the manufacturer's instructions.

### Statistical analysis

The quantitative data are expressed as means ± standard error of mean (SEM). Statistical analyses were performed by unpaired t test with GraphPad Prism 5 software (San Diego, Ca, USA). A value of $p < 0.05$ was considered statistically significant.

## Results

### Comparison of keratinocyte and fibroblast growth and proliferation on flat and nanopattern

For the previous reports that growth and proliferation of cells depend on substrate surface modifications, the epithelial and mesenchymal lineages, keratinocytes and fibroblasts were seeded at a cell density of $1 \times 10^5$ per 35 mm dish on both flat and nanopattern substrates. Each pore of the nanopattern had a diameter of 200 nm and a depth of 500 nm with a distance of 500 nm between the centers of the pores (Fig 1A). After attachment of the cells, each cell growth was observed for 72 hours using a live microscope. Fibroblasts showed slow growth on the nanopore surface, whereas keratinocytes were not affected by the nanopore surface. At day 1, both cell types adopted a normal morphology and were uniformly distributed across both surfaces. This pattern of behavior persisted until 3 days when a clear difference emerged in fibroblasts as the flat group adopted a proliferation whereas a limited growth was shown in fibroblasts cultivated on the nanopattern substrate. In contrast, keratinocytes continued to proliferate and grow with little to no visible difference between both groups (Fig 1B and S1–S4 Movies).

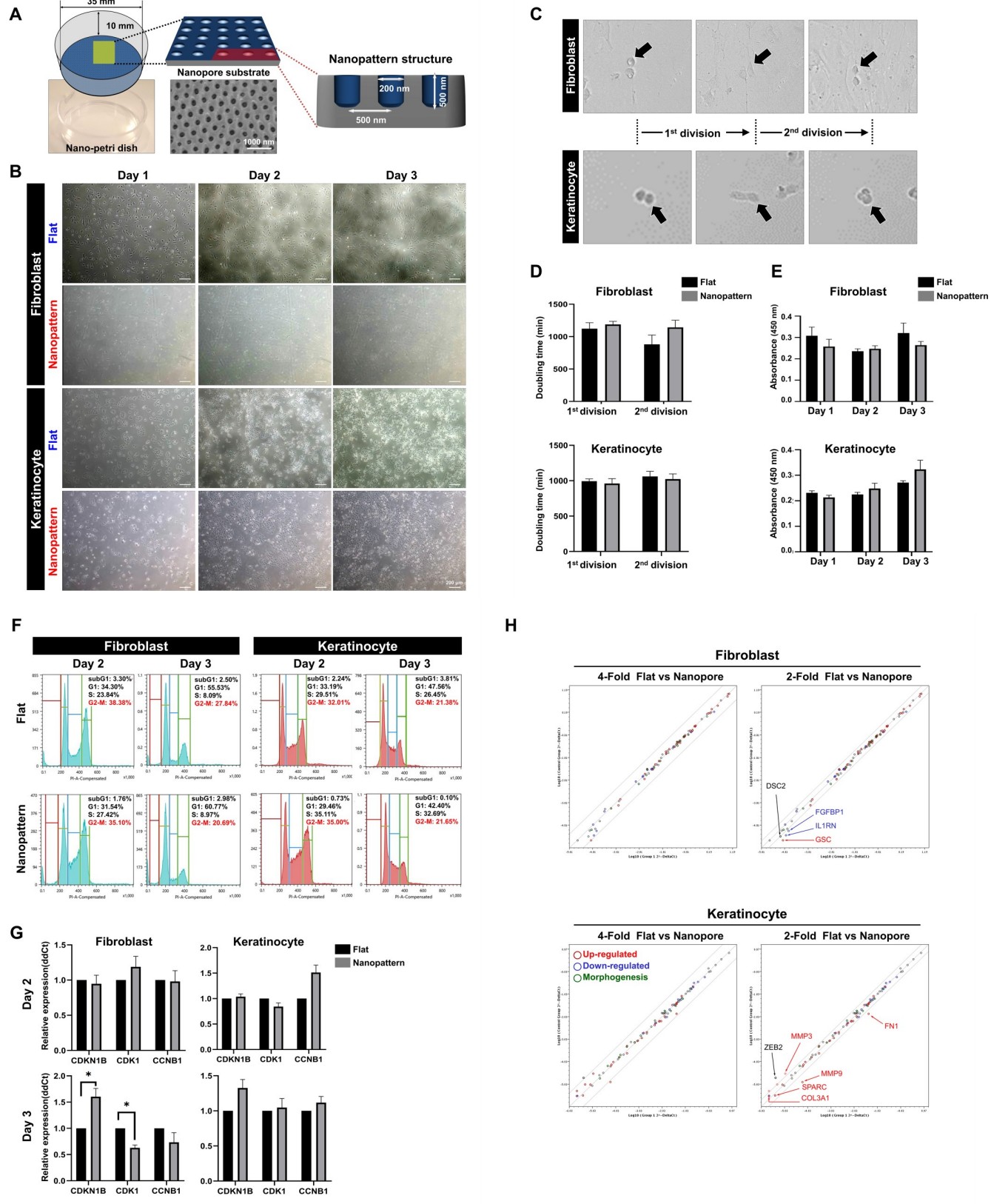

**Fig 1. Nanopattern as a topographical tool for the inhibition and mediation of fibroblast growth.** (A) Design and parameters (depth, diameter, and layout) of nanopattern. (B) Growth and proliferation of keratinocytes and fibroblasts recorded. (C) Mitosis of keratinocytes and fibroblasts on nanopatterns observed by microscopy. (D) Quantification of doubling time (proliferation) of keratinocytes and fibroblasts. (E) Quantification of absorbance (growth) of keratinocytes and fibroblasts. (F) Cell cycle analysis of populations entering the G1, S, or G2-M phase at day 2 and 3. (G) Expression levels of cell cycle genes CDKN1B, CDK1, and CCNB1 at day 2 and 3. (H) Scatter plot analysis of differential gene expression for EMT-related genes.

In order to more accurately analyze the phenomenon of growth observed differently in the pattern, we followed five or more dividing cells with live image analysis (Fig 1C). As a result, we found that the doubling time of keratinocytes was similar for the first and second division cycle for both substrates but fibroblasts exhibited a protracted duration for the second division cycle when cultivated on the nanopatterned dishes, (Fig 1D). Furthermore, fibroblast mobility appeared to be hindered in the nanopattern group whereas cells in the flat group were highly active throughout the cultivation process (S1 and S2 Movies). When CCK-8 analysis was conducted to further quantify the proliferation rate of each group, the absorbance value of fibroblasts was lower for the nanopattern group and remained comparatively inert at each time point while keratinocytes steadily increased over time (Fig 1E).

Based on these observations, to further investigate the proliferative state of each group, cell cycle analysis was performed by propidium iodide (PI) staining to quantify the percentage of cells in the division phase at each respective time; day 2 and day 3. As a result, we determined that the percentage of fibroblasts in the G2-M phase was exhibited larger differences as time progressed when comparing the flat and nanopattern group at day 2 (38.38% vs. 35.10%) and day 3 (27.84% vs. 20.69%). However, keratinocytes showed similar at day 2 (32.01% vs. 35.0%) and day 3 (21.38% vs. 21.65%) for both the flat and nanopattern group, respectively (Fig 1F). Next, the expression of key genes involved in cell cycle progression were examined to investigate whether the change in fibroblasts growth and proliferation were reflected at the genetic level. We found that cyclin-dependent kinase inhibitor 1B (CDKNB1) which controls cell cycle progression at the G1 phase by inhibiting cell division [22], was upregulated in fibroblasts at day 3 for the nanopattern group while its expression remained static for the flat group. In tandem, cyclin dependent kinase 1 (CDK1) and the regulatory protein for mitosis, cyclin B1 (CCNB1) [23], was markedly downregulated at day 3 for the nanopattern group, suggesting the inhibition of proliferation (Fig 1G). In contrast, CCNB1 was upregulated in keratinocytes at day 2 and its elevated expression persisted at day 3 in conjunction with normal levels of CDK1 expression, which is indicative of proliferative cells as the cyclin B1-cdk1 complex is involved in the early events of mitosis [24]. In addition, we investigated the effect of nanopore substrates on the epithelial-mesenchymal transition (EMT) of keratinocyte and fibroblast. A customized EMT array was employed as a means to examine differential gene expression for 84 key genes that are specific but not limited to EMT and its associated processes. As such, the array includes a multitude of genes related to the extracellular matrix, cell morphogenesis, development, growth, and proliferation as well as signal transduction and transcription factors. A scatter plot analysis of the genes exhibiting a 2 or 4-fold change revealed that the majority remained unchanged, confirming that nanopore substrates had little to no effect on the transformation of keratinocytes and fibroblasts (Fig 1H). Furthermore, genes that were upregulated by at least 2-fold in fibroblasts such as COL3A1, MMP3, and MMP9 are pertinent to matrix remodeling while genes that were downregulated by at least 2-fold such as FGFBP1 are associated with fibroblast growth [13]. These results show that nanopore substrates do not alter the state of both cells but rather inhibit growth related factors required for expansion which is consistent with our microscopic observations. Taken together, these results demonstrate that epithelial lineage cell such as keratinocytes proliferate at a similar rate on nanopatterns relative to flat surfaces but mesenchymal lineage cell such as fibroblasts exhibit reduced proliferation.

## Analysis of co-culture potential for epithelial lineage cells and mesenchymal lineage cells on nanopatterned dishes

From the results that cell growth can be regulated by nanopatterned culture dishes, we hypothesized that nanopatterning could be used for co-culture microenvironments that can inhibit the growth of mesenchymal lineage cells while maintaining the growth of epithelial lineage cells. Since the nanopatterning had no significant effect on the growth and proliferation of epithelial lineage cell; keratinocytes but hindered the proliferative capacity of mesenchymal lineage cell; fibroblasts, we decided to co-cultivate human pluripotent stem cells (hPSCs) which are also of the epithelial lineage, on the mouse fibroblast in nanopatterned environments (Fig 2A). To analyze whether nanopatterned dishes were effective in co-culture of mesenchymal lineage cells and epithelial lineage cells, we observed mouse fibroblast cells and hPSCs co-cultured for 3 days on flat culture plates and nanopatterned culture dishes. At the hPSCs with mouse fibroblast density of $1 \times 10^5$, we observed that mouse fibroblasts began to overlay hPSCs within 3 days for the flat group whereas hPSCs on nanopatterns were capable of retaining its colonial jurisdiction while expanding in a compact, well-rounded and size sufficient manner (Fig 2B, white arrow). This was clearly visible in live imaging which also revealed mouse fibroblasts inactivity arising earlier for the flat group followed by hPSCs extrication whereas the nanopattern group displayed prolonged mouse fibroblasts activity and retained hPSCs—mouse fibroblasts interaction (S5 and S6 Movies).

Since mouse fibroblasts overlay the hPSCs in flat dish, co-culture is usually performed by treatment with mitomycin-c (MMC) that inhibits mouse fibroblast growth. For this reason, we further investigated the maintenance potential of hPSCs co-cultivated with proliferative mouse fibroblasts on the nanopattern through multiple passages in comparison to MMC-treated (with MMC) and non-treated (w/o MMC) groups on flat surfaces. As expected, MMC-treated mouse fibroblasts were capable of proliferating hPSCs for 4 days, whereas non-treated groups were incapable of growing due to proliferative mouse fibroblasts. In contrast, proliferative mouse fibroblasts on nanopatterns were able to support hPSCs expansion for 4 days and across 5 passages as the colony expanded at a similar rate to that of the MMC-treated group (Fig 2C, white circle). Quantification of colony size (white circles in Fig 2C) was performed based on its diameter which further supported our observations as the diameter of the hPSC colony was measured to range between 900 μm to 1250 μm depending on its passage (Fig 2D). Next to compare the characteristics of maintained hPSCs for 5 passages in proliferative- and MMC-treated mouse fibroblasts, RT-PCR was conducted for representative pluripotent markers such as OCT4, SOX2, and E-cadherin, revealing similar levels of OCT4 and SOX2 expression but intensified E-cadherin expression in the nanopattern group (Fig 2E). This was further confirmed by immunocytochemistry for OCT4 and E-cadherin, revealing protein expression in a pattern similar to that of gene expressions. Interestingly, gene and protein expression analysis showed that E-cadherin expression was less compact in the flat group than in the nanopattern group (Fig 2F). In general, E-cadherins are found in epithelial tissue as a type of cell-cell adhesion molecule that is important in the formation of adherens junctions to bind cells with each other. Unlike hPSCs that easily detach in 3 days in flat structure (S5 Movie), they adhere well for 5 days and can be passaged on the nanostructures, which can be interpreted as an improvement in the adhesion ability of hPSCs due to the increased expression of e-cadherin. Although the size of hPSC colonies was dependent on the density of proliferative mouse fibroblasts, it was confirmed by alkaline-phosphatase (AP) staining that they were well maintained for 5 days without the detachment inclination of undifferentiated hPSC colonies (S1 Fig). From these results, the high expression of E-cadherin in the epithelial phenotype cells and the hPSC cultured on nanopatterned dishes, suggests that the pattern structure enhances the adhesion

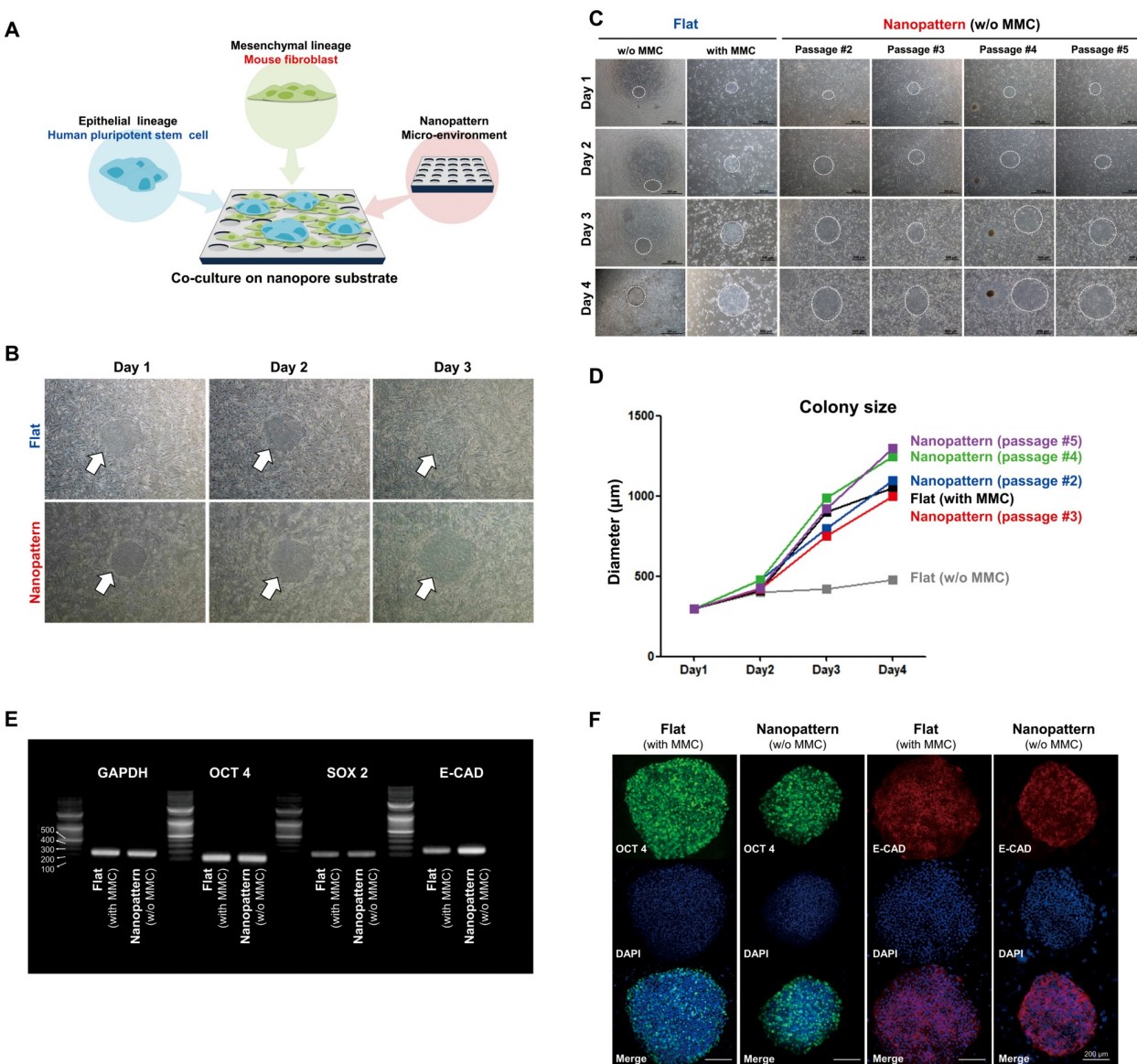

**Fig 2. Analysis of co-culture potential of epithelial lineage cells and mesenchymal lineage cells using nanopatterns.** (A) Schematic diagram of the development strategy for co-culture system using nanopatterned dishes. (B) Expansion of hPSCs co-cultured with proliferative mouse fibroblasts on flat and nanopattern using for 3 days. (C) Observation of growth capacity for hPSCs co-cultured with proliferative fibroblast and MMC-treated fibroblast on flat and nanopattern throughout multiple passages. (D) Diametric quantification of colony size for Fig 2C. (E and F) Comparison of the pluripotent specific genes (E) and proteins (F) expression between hPSCs co-cultured with MMC-treated fibroblast cells on flat and hPSCs co-cultured with proliferative (without MMC treatment) fibroblast cells on nanopatterned dishes.

ability of the OCT4-expressing undifferentiated hPSCs (Fig 2). We also examined the differential gene expression of 84 key genes that are specific but not limited to the Epithelial-Mesenchymal Transition (EMT) program for 5-passaged hPSCs to confirm that the cells sustain their pluripotent character. A scatter plot analysis of the genes exhibiting a 2-fold change revealed that the majority of genes involved in EMT or differentiation remained unchanged, confirming that nanopatterns had little to no effect on the undifferentiated characteristic and pluripotency of hPSCs (S2 Fig). Collectively, these results demonstrate sustainable hPSC cultivation with proliferative mouse fibroblasts by utilizing nanopatterns which also improved colonial integrity in comparison to flat substrates.

## Analysis of cell secretomes profiles for MMC-treated fibroblasts in flat and proliferative fibroblasts on nanopatterned dishes

We thought that the growth of fibroblasts with delayed growth by micro-environmental change on nanopatterned substrates was different from those of chemically inhibited growth by MMC treatment. So, we tried to identify the difference in cell activity by analyzing the secretomes and metabolism of fibroblast cells cultured under different conditions (Fig 3A). First, to identify and investigate the multitude of cytokines and chemokines secreted from cultured fibroblasts for 5 days that induce important paracrine effects, densitometry-based cytokine array analysis was performed for both the MMC-treatment group and nanopattern group. As a result, a total of 51 and 59 cytokines were successfully detected and quantified in MMC-treatment and nanopatterning, respectively. Unexpectedly, it was confirmed that more cytokines was secreted from the fibroblasts that were inhibited by MMC treatment than the growth delayed fibroblasts on nanopatterned dishes (Fig 3B). A heat map was generated from densitometry based quantifications of all 62 cytokines expressed in both groups to better visualize the change in expression for each proteomic candidate which revealed several specific as well as non-specific factors with exponential change (Fig 3C). 62 cytokines can be classified into five cell activation groups such as ECM remodeling, differentiation, adhesion, viability and immune regulation, among which 18 cytokines differed more than four-fold (Fig 3C, letters in red). Of the 18 cytokines, 13 cytokines were secreted more than four times in MMC-treated fibroblasts, and interestingly, the secretion of VEGF, DKK-1 and HGF factors involved in cell differentiation was more than seven times higher (Fig 3D). In the case of fibroblasts cultured on nanopatterned dishes, 5 cytokines including cell adhesion cytokine JAM-A, were secreted at least 10-fold. Especially cell viability-related IGF-II, OPG, and IL6 cytokines were significantly secreted over 15-fold (Fig 3E). Besides, 19 cytokines secreted more than four-fold in both groups were mostly cell activators involved in cell proliferation and survival as expected (Fig 3F). This indicates that fibroblasts secrete a lot of cell activating factors and can help the survival and proliferation of co-cultured cells such as hPSCs. However, we found that the type and concentration of cytokines secreted by fibroblasts differed according to growth inhibition by MMC treatment or growth delay by micro-environmental regulation by nanopatterning. Furthermore, proliferative fibroblasts showed higher proteome concentrations for cell survival and proliferation-related factors than growth arrested fibroblasts. According to previous studies, differentiation and growth activity modulators such as VEGF, DKK-1 and HGF factors can be increased by cell death [25, 26]. So the high levels of VEGF, DKK-1 and HGF secretion in MMC-treated fibroblasts could be inferred by cells death. When co-culture of MMC-treated fibroblasts and hPSCs sub-cultured at 5-day intervals for more than 5 days, the death of fibroblasts and the differentiation of hPSCs can be observed simultaneously (S3 Fig). Therefore, MMC-treated fibroblasts need to be renewed every 5 days for the maintenance of undifferentiated hPSCs. These results suggest that chemically inhibited cell growth may alter the ability of cell activation factor secretion by accelerating cell death, while delayed cell growth by micro-environment control could maintain more stable cell activation factor secretion ability.

## Comparison of metabolic ability between proliferative mouse fibroblasts and MMC-treated mouse fibroblasts

Unlike the micro-environment control, inhibition of cell growth by chemical treatment methods accelerates cell death [27]. Since cell death also affects the metabolic capacity of cells, we analyzed the metabolism change of fibroblasts by MMC treatment, a chemical cell growth inhibition method. To measure the change in metabolic capacity of mouse fibroblasts

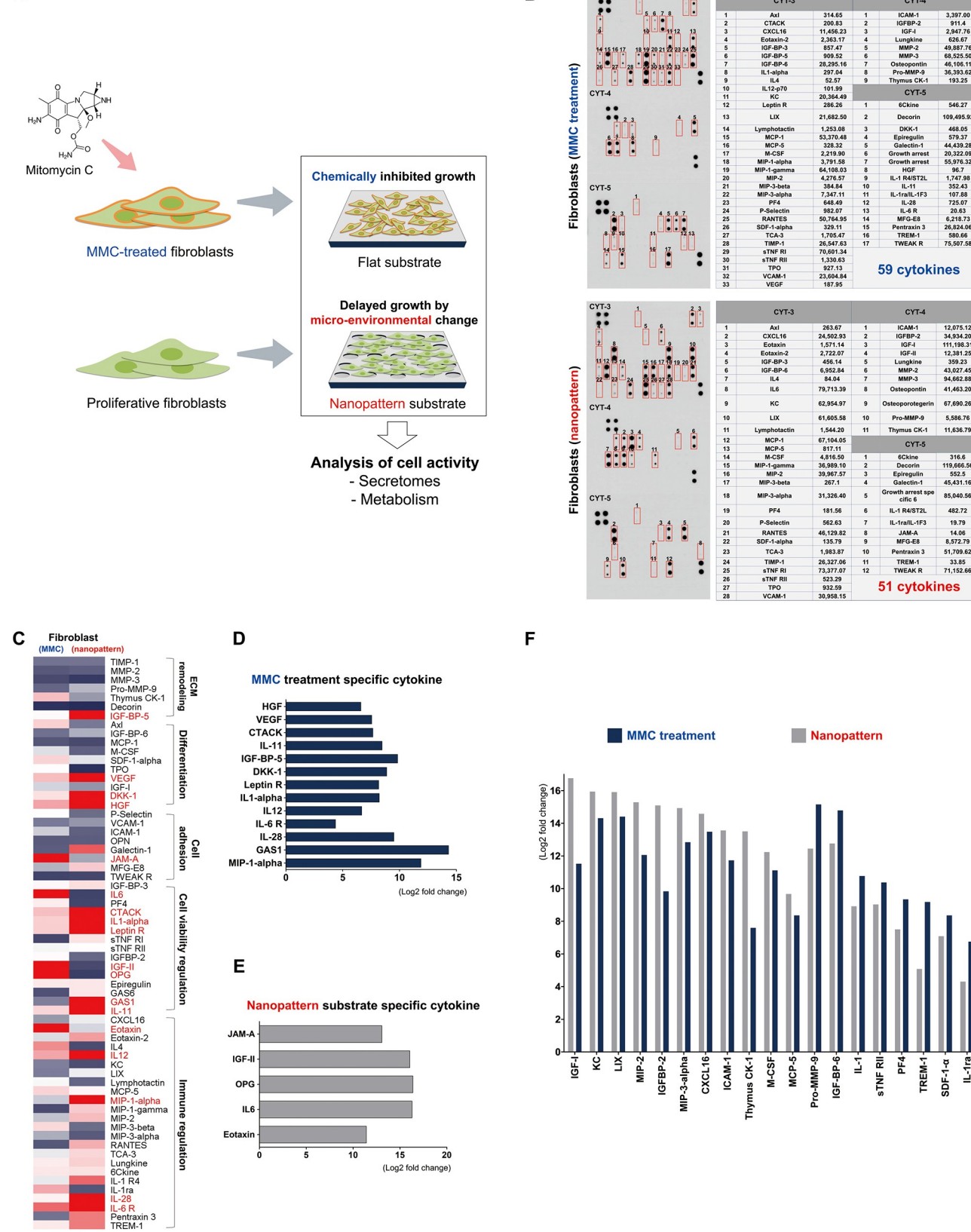

**Fig 3. Analysis of cell secretomes for MMC-treated and proliferative fibroblasts.** (A) Experimental Schematic of Comparison of Cell Activity on Flat and Nanopatterned Substrates (B) Densitometry based quantification of identified secretomes. (C) Heat map depicting fold change values for each candidate. (D and E) Specific factors secreted by MMC-treated and proliferative fibroblasts. (F) Fold change values for putative factors expressed by both groups.

according to MMC concentration, we investigated the effect of MMC on the Oxygen Consumption Rate (OCR) and Extracellular Acidification Rate (ECAR) of fibroblasts after administering 10 μg/mL or 20 μg/mL of MMC. In comparison to the control, the MMC-treated groups showed declining values for OCR in a concentration dependent manner. This reduction was also reflected in the basal and maximal respiration values as well as ATP production following quantitative analysis (Fig 4A). Similarly, ECAR values declined with increasing MMC concentration, suggesting that glycolysis is also hindered by MMC treatment (Fig 4B). These results are indicative of abnormal metabolic rates in growth arrested fibroblasts by MMC treatment which is likely to disrupt the management of biochemical reactions that are essential for cellular maintenance. Compared to cells cultured in nanopatterns, the metabolic capacity of fibroblasts decreased with MMC treatment concentration. Unfortunately, the use of nanopatterned metabolic assay plates could not be used to compare direct metabolic capacity changes. Although direct comparisons were difficult, MMC treatment reduced the metabolic capacity of the cells compared to the nanopatterns because fibroblasts cultured in the nanopatterns were used as a control for the accuracy of the experiment.

## Discussion

The development of cell culture technology using nanopatterning similar to the surface structure of *in vivo* tissues has been progressing. In this study, we found that nanopatterned structures delayed the growth of mesenchymal type cells by reducing the expression of G2-M stage-related genes in cell cycle, while promoting the attachment and growth of epithelial type cells by enhancing the expression of adhesion proteins in the nanopattern. In general, a method of co-culture with mesenchymal type cells that secrete various growth factors is used for the growth of epithelial type cells that are difficult to culture. At this time, mesenchymal type cells with rapid cell growth are used for co-culture by inhibiting cell growth with chemicals or ultraviolet. Herein, we applied micro-environment substrate to the co-culture of epithelial type and mesenchymal type cells using the characteristics of our nanopatterned structures, which have different growth responses depending on the cell type.

Most tissues in the body consist of a combination of epithelial and mesenchymal cells. In order to mimic the tissue composition in body, many researchers experimented with co-culture of epithelial and mesenchymal type cells [28, 29]. Representative epithelial and mesenchymal type cell co-culture methods are used for astrocytes and neurons, melanocytes and fibroblasts and hPSC and fibroblasts [30–32]. However, because mesenchymal cells having a large surface area and rapid growth rate inhibit the growth of epithelial cells, they are used for co-culture by inhibiting the growth of mesenchymal cells by ultraviolet and chemical treatment (Fig 2B). Inhibiting the growth of cells by ultraviolet and chemical treatment reduces the function of the cells due to a decrease in the activity and metabolic capacity of the cells, which eventually promotes cell death [33–36]. In this study, we found that micro-environmental changes using nanopatterned structures that can regulate cell surface contact can delay mesenchymal cells growth and promote epidermal cells growth (Fig 1). These findings were confirmed by co-culture with epithelial type cells; hPSC and mesenchymal type cells; fibroblasts, and the results of this study suggest a new co-culture method without ultraviolet and chemical treatment (Fig 5).

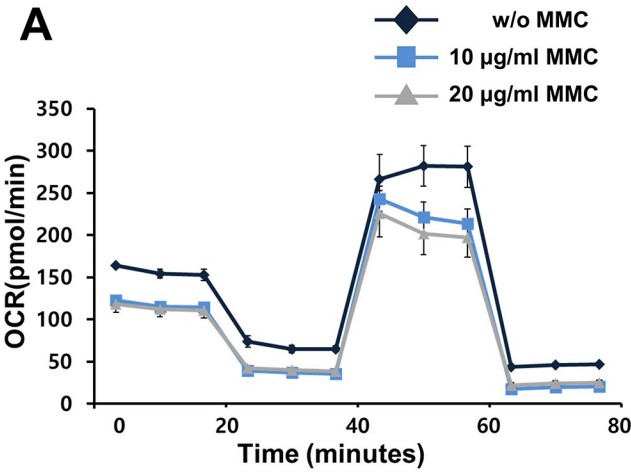

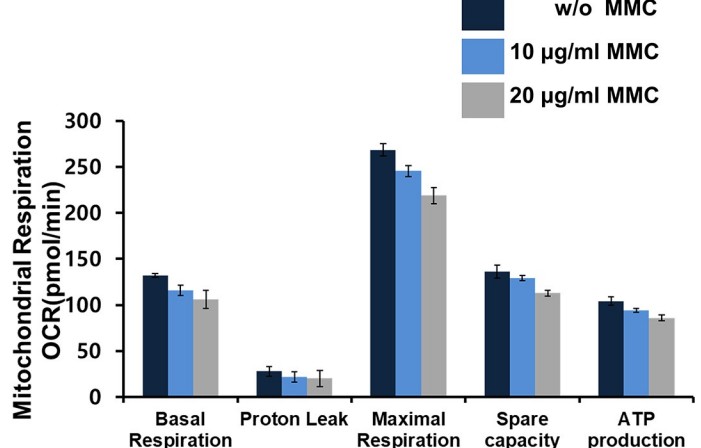

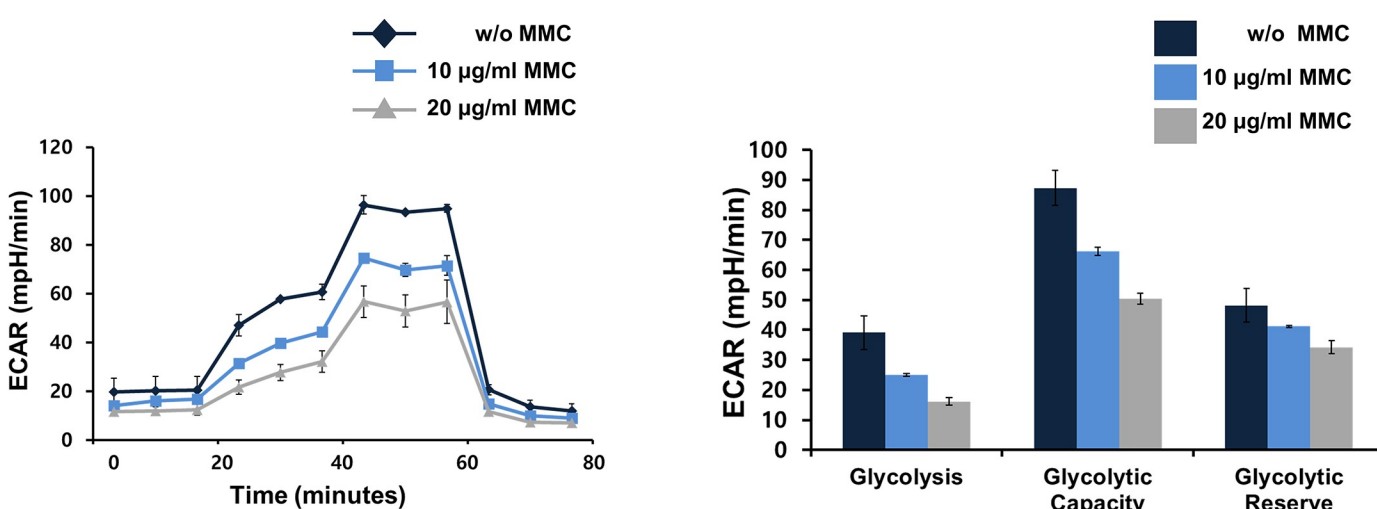

**Fig 4. Comparison of metabolic ability with MMC treatment concentration.** (A) Oxygen Consumption Rate (OCR) and (B) Extracellular Acidification Rate (ECAR) of fibroblasts treated with 0, 10, and 20 µg/ml. In the case of 0 µg/ml (without) treated group, fibroblasts cultured in the nanopatterns were used as a control group.

The primary purpose of epithelial and mesenchymal cells co-culture to mimic tissue structure is to effectively cultivate epithelial cells that are difficult to maintain and proliferate *in vitro* using various growth factors secreted from mesenchymal type cells. However, inhibiting the growth of mesenchymal cells by chemical treatment for co-culture promotes cell death due to growth inhibition, and the substances secreted during cell death may adversely affect

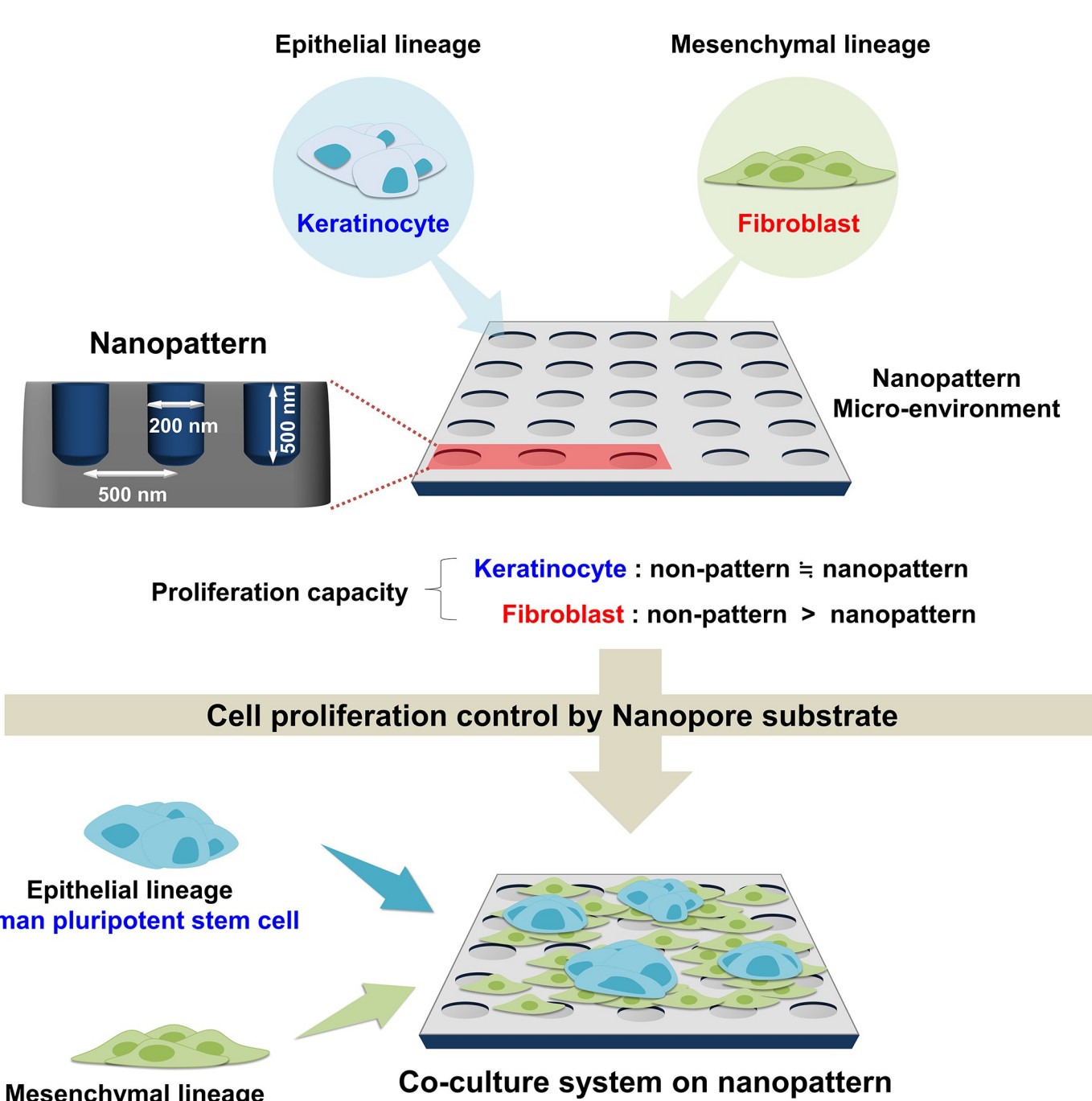

**Fig 5. Schematic diagram of a new coculture system strategy using nanopatterns.**

epithelial type cells. Fig 3D results show that the growth inhibited fibroblasts by MMC-treatment secrete the various growth factors such as VEGF, HDF, and DKK1, which induce differentiation of hPSCs. Also, hPSCs co-cultured with MMC-treated fibroblasts for more than 5 days could be observed to promote differentiation (S3 Fig). Therefore, depending on the

management of MMC-treated fibroblasts, effective undifferentiated hPSC culture objectives may be achieved or, in contrast, induction of hPSCs differentiation.

Unlike MMC treatment, growth delay of mesenchymal type cells through micro-environmental control does not affect cell death, which results in reduced secretion of differentiation-related factors by cell death while the secretion of positive factors for the maintenance of undifferentiated hPSC was increased (Figs 2 and 3). In Fig 4, cell growth was inhibited by chemical treatment, which was confirmed that the metabolic capacity of the cells was significantly decreased according to the concentration of MMC in the process of cell death. From these results, the inhibition of cell growth by chemical methods has a limitation as a co-culture method as it decreases cell activity, metabolism, and both function of cells while promoting cell death. On the other hand, co-culture using nanopatterned dishes selectively delays the growth of only the mesenchymal type cells, and does not inhibit cell activity, metabolism or induce cell death. In addition, the nanopattern structure is similar to the surface modification of tissues in the body and is effective in the attachment of epithelial type endothelial cells [37], and also we have previously reported that epithelial type undifferentiated hESC proliferation and pancreatic cell differentiation efficiency were improved [13, 38].

In this study, we also confirmed that the adhesion of epithelial type hPSCs in the nanopattern structures is enhanced and has an effect of promoting growth. In Fig 2E and 2F, the expression of e-cadherin was enhanced in nanopatterns, and epithelial type undifferentiated hPSCs were cultured for 5 days in different cell densities of proliferative mouse fibroblasts for evaluation of cell adhesion capacity on nanopattern structures. In S1 Fig, hPSCs showed a large colony size at low density, whereas small colonies were grown at high density, and alkaline-phosphatase (AP) staining analysis showed that the undifferentiation state was well maintained. Interestingly, hPSC cells in $1 \times 10^5$ proliferative mouse fibroblast density dropped in 3 days in flat (S5 Movie), whereas hPSC cells incubated for 5 days in $1.5 \times 10^5$ proliferative mouse fibroblast density in nanopatterns (S1 Fig), which could be concluded that the enhanced e-cadherin expression of hPSC cells on the nanopatterned dishes increased the cell adhesion ability. As such, micro-environment regulation by nanopatterns improves the attachment and growth of epithelial type cells and reduces mesenchymal type cells growth. Thus, mesenchymal type cells such as fibroblasts, astrocytes and mesenchymal stem cells that secrete various growth factors can easily be co-cultured with epithelial type cells such as melanocytes, neurons and undifferentiated hPSCs that are difficult to proliferate without complex processes. Taken together, the results of this study suggest that nanopatterning can be used as a new technique without having to be concern about the problems such as decreased activity, metabolic capacity and accelerated cell death caused by chemical MMC treatment for the inhibition of mesenchymal type cell growth. Although the current nanopatterning process is both imprecise and expensive, it is expected that it will be used in various cell culture techniques including co-culture if technology development and mass production systems are established in the future.

## Supporting information

**S1 Fig. Comparison of the undifferentiated hPSC colony size on the various density of proliferative mouse fibroblasts.**
(TIF)

**S2 Fig. Scatter plot analysis of differential gene expression for pluripotency and EMT-related genes.**
(TIF)

**S3 Fig. Morphological changes in hPSCs co-cultured with MMC-treated fibroblasts over time of incubation.**
(TIF)

**S1 Table. Primer list of real-time PCR.**
(TIF)

**S2 Table. Primer list of reverse transcription PCR.**
(TIF)

**S1 Movie. Fibroblast culture on flat dish.**
(AVI)

**S2 Movie. Fibroblast culture on nanopattern dish.**
(AVI)

**S3 Movie. Keratinocyte culture on flat dish.**
(AVI)

**S4 Movie. Keratinocyte culture on nanopattern dish.**
(AVI)

**S5 Movie. Co-culture on flat dish.**
(AVI)

**S6 Movie. Co-culture on nanopattern dish.**
(AVI)

**S1 Raw images.**
(PDF)

## Author Contributions

**Conceptualization:** Kang Moo Huh, Dong Sung Kim, Sun-Woong Kang, Sung-Hwan Moon.

**Data curation:** Taek-Hee Jung, Eun-Bin Chung, Hyung Woo Kim, Seong Woo Choi.

**Formal analysis:** Taek-Hee Jung, Eun-Bin Chung, Soon-Jung Park, Anthony Safaa Mukhtar, Eunmi Kim.

**Methodology:** Soon-Jung Park, Anthony Safaa Mukhtar, Eunmi Kim.

**Resources:** Hyung-Min Chung.

**Supervision:** Hyung-Min Chung, Kang Moo Huh, Dong Sung Kim, Sun-Woong Kang, Sung-Hwan Moon.

**Writing – original draft:** Hyung Woo Kim, Seong Woo Choi, Sung-Hwan Moon.

**Writing – review & editing:** Dong Sung Kim, Sun-Woong Kang.

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
