## [Decision Letter · Decision Letter 0]

1 Apr 2020

PONE-D-19-35930

Application of co-culture technology of epithelial type cells and mesenchymal type cells using nanopatterned structures

PLOS ONE

Dear Dr Moon,

Thank you for submitting your manuscript to PLOS ONE. After careful consideration, we feel that it has merit but does not fully meet PLOS ONE’s publication criteria as it currently stands. Therefore, we invite you to submit a revised version of the manuscript that addresses the points raised during the review process.

We would appreciate receiving your revised manuscript by May 16 2020 11:59PM. To enhance the reproducibility of your results, we recommend that if applicable you deposit your laboratory protocols in protocols.io, where a protocol can be assigned its own identifier (DOI) such that it can be cited independently in the future. For instructions see: http://journals.plos.org/plosone/s/submission-guidelines#loc-laboratory-protocols

We look forward to receiving your revised manuscript.

Kind regards,

Xuefeng Liu

Academic Editor

PLOS ONE

Journal Requirements:

1. Thank you for including your competing interests statement; "The authors have declared that no competing interests exist."

We note that one or more of the authors are employed by a commercial company: "T&R Biofab Co. Ltd"; and "Amorepacific Corporation"

3.

PLOS ONE now requires that authors provide the original uncropped and unadjusted images underlying all blot or gel results reported in a submission’s figures or Supporting Information files. This policy and the journal’s other requirements for blot/gel reporting and figure preparation are described in detail at https://journals.plos.org/plosone/s/figures#loc-blot-and-gel-reporting-requirements and https://journals.plos.org/plosone/s/figures#loc-preparing-figures-from-image-files. When you submit your revised manuscript, please ensure that your figures adhere fully to these guidelines and provide the original underlying images for all blot or gel data reported in your submission. See the following link for instructions on providing the original image data: https://journals.plos.org/plosone/s/figures#loc-original-images-for-blots-and-gels.

Reviewers' comments:

Reviewer's Responses to Questions

**Comments to the Author**

1. Is the manuscript technically sound, and do the data support the conclusions?

Reviewer #1: Yes

2. Has the statistical analysis been performed appropriately and rigorously? 

Reviewer #1: Yes

3. Have the authors made all data underlying the findings in their manuscript fully available?

Reviewer #1: Yes

4. Is the manuscript presented in an intelligible fashion and written in standard English?

Reviewer #1: Yes

5. Review Comments to the Author

Reviewer #1: This study firstly used the nanopattern technology to co-culture mesenchymal cells and epithelial cells. They found that the nanopattern substrates could inhibit the proliferative rate of fibroblasts, but do not influence the growth of keratocytes. Importantly, the application of this technology did not affect the metabolism ability of fibroblasts. They also performed molecule mechanisms underlying these observations. They found that more fibroblasts were arrested in G1 phase and the gene expression of CDKNB1 was upregulated in nanopatterned dishes, comparing to the flat dishes.

Overall, the article is completed and justified. If the authors can carefully revise the English writing throughout the paper and submit high-quality pictures, this manuscript can be considered to be accepted.

6. PLOS authors have the option to publish the peer review history of their article (what does this mean?). If published, this will include your full peer review and any attached files.

Reviewer #1: No

---

## [Author Response · Author response to Decision Letter 0]

13 Apr 2020

Response to Journal & Reviewer 

Manuscript ID: PONE-D-19-35930 

Manuscript Title: Application of co-culture technology of epithelial type cells and mesenchymal type cells using nanopatterned structures 

The authors thank the editor and the reviewer for thoughtful comments.

Journal Requirements: 

1. Thank you for including your competing interests statement; "The authors have declared that no competing interests exist." We note that one or more of the authors are employed by a commercial company: "T&R Biofab Co. Ltd"; and "Amorepacific Corporation"

1.1 Funding 

Please provide an amended Funding Statement declaring this commercial affiliation, as well as a statement regarding the Role of Funders in your study.

1.2 Competing Interests

Please also provide an updated Competing Interests Statement declaring this commercial affiliation along with any other relevant declarations relating to employment, consultancy, patents, products in development, or marketed products, etc.

Answer: T&R Biofab Co. Ltd and Amorepacific Corporation did not provide financial support, and did not have any roles in this study. The authors belonging to T&R Biofab and Amorepacific companies recently moved to the companies, and no funds have been contributed by the companies. 

1.1 Funding

This research was supported by the Bio & Medical Technology Development Program of the National Research Foundation funded by the MSIP (NRF-2016M3A9B4919616, NRF-2019M3A9H1103331) and by a grant (20000325) from the Technology Innovation Program funded from the Ministry of Trade, Industry and Energy (MOTIE), and National Research Foundation grant (MSIT) (No. 2017R1A2A1A05001090) funded by the Republic of Korea government. 

1.2 Competing Interests

The authors have declared that no competing interests exist. T&R Biofab Co. Ltd and Amorepacific Corporation do not alter our adherence to PLOS ONE policies on sharing data and materials.

2. PLOS requires an ORCID iD for the corresponding author in Editorial Manager.

Answer: The corresponding author updated the ORCID iD in online system.

3. PLOS ONE now requires that authors provide the original uncropped and unadjusted images underlying all blot or gel results reported in a submission’s figures or Supporting Information files.

Answer: Original images of Figure 2E were uploaded through ‘Fig 2E_raw_images’.

Reviewer Comments: 

Reviewer #1: This study firstly used the nanopattern technology to co-culture mesenchymal cells and epithelial cells. They found that the nanopattern substrates could inhibit the proliferative rate of fibroblasts, but do not influence the growth of keratocytes. Importantly, the application of this technology did not affect the metabolism ability of fibroblasts. They also performed molecule mechanisms underlying these observations. They found that more fibroblasts were arrested in G1 phase and the gene expression of CDKNB1 was upregulated in nanopatterned dishes, comparing to the flat dishes.

Overall, the article is completed and justified. If the authors can carefully revise the English writing throughout the paper and submit high-quality pictures, this manuscript can be considered to be accepted.

Answer: We updated the high quality images, and English writing was revised by a native speaker.

---

## [Editor Report · Decision Letter 1]

24 Apr 2020

Application of co-culture technology of epithelial type cells and mesenchymal type cells using nanopatterned structures

PONE-D-19-35930R1

Dear Dr. Moon,

We are pleased to inform you that your manuscript has been judged scientifically suitable for publication and will be formally accepted for publication once it complies with all outstanding technical requirements.

With kind regards,

Xuefeng Liu

Academic Editor

PLOS ONE
---

## [Editor Report · Acceptance letter]

29 Apr 2020

PONE-D-19-35930R1 

Application of co-culture technology of epithelial type cells and mesenchymal type cells using nanopatterned structures 

Dear Dr. Moon:

I am pleased to inform you that your manuscript has been deemed suitable for publication in PLOS ONE. Congratulations! Your manuscript is now with our production department. 

With kind regards,

on behalf of

Dr. Xuefeng Liu 

Academic Editor

PLOS ONE